

# New Global Mean Dynamic Topography CNES-CLS-22 Combining Drifters, Hydrography Profiles and High Frequency Radar Data

Solène Jousset[1], Sandrine Mulet[1], Eric Greiner[1], John Wilkin[2], Lien Vidar[3], Léon Chafik[4], Roshin Raj[5], Antonio Bonaduce[5], Nicolas Picot[6], Gérald Dibarboure[6]

[1]CLS, Ramonville Saint Agne, 31250, France
[2]Institute of Marine and Coastal Sciences, Rutgers University, New Brunswick, New Jersey, USA
[3]Institute of Marine Research, Bergen, Norway
[4]Department of Meteorology and Bolin Centre for Climate Research, Stockholm University, Stockholm, Sweden
[5]Nansen Environmental and Remote Sensing Center, Bjerknes Centre for Climate Research, Bergen, Norway
[6]CNES, Ramonville Saint Agne, 31520, France

*Correspondence to*: Solène Jousset (sjousset@groupcls.com)

**Abstract.** The mean dynamic topography (MDT) is a key reference surface for altimetry. It is needed for the calculation of the ocean absolute dynamic topography, and under the geostrophic approximation, the estimation of surface currents. CNES-CLS MDT solutions are calculated by merging information from altimeter data, GRACE, and GOCE gravity field and oceanographic in situ measurements (drifting buoy velocities, High Frequency radar surface velocities, hydrological profiles).

The objective of this paper is to present the newly updated CNES-CLS22 MDT. The main improvement of this new CNES-CLS22 MDT over the previous CNES-CLS18 MDT is in the Arctic, with better coverage and a more physical solution (with the disappearance of artifacts from the previous version). This is due to the use of a new first guess estimated with the CNES-CLS22 MSS and the GOCO06s geoid to which optimal filtering has been applied, as well as Lagrangian filtering at the coast to reduce the intensity of currents normal to the coast. Improvements also include updating the drifting buoy and T/S profile

databases, and processing to obtain synthetic mean geostrophic velocities and synthetic mean heights. In addition, a new data type, HF radar data, was processed to extract physical content consistent with MDT in the Mid Atlantic Bight of the northeast U.S. coastal region. The study of this region in particular has shown the improvements of the CNES-CLS22 MDT, though there is still work to be done to obtain a more physical solution over the continental shelf. The CNES-CLS22 MDT has been evaluated against independent height and velocity data in comparison with the previous version, the CNES-CLS18. The new

solution presents slightly better results, although not identical in all regions of the globe.

## 1 Introduction

Since the early days of altimetry, estimating absolute dynamic topography (ADT) accurately has been a challenge (Rio 2010). The ability to reconstruct absolute dynamic topography at the resolution of along-track altimetry, i.e. 7 km at 1 Hz and 300 m

at 20Hz, is limited by the accuracy of the geoid at these scales. So most scientific studies of the ocean have used the anomaly of sea level relative to the temporal Mean Sea Surface (MSS) computed over a long reference period: the sea level anomaly





(SLA). The dynamics of mesoscale structures are readily apparent in SLA, but many scientific and operational activities require accurate absolute sea level that includes the contribution from MDT. For the study of eddy-mean interactions, a positive sea level anomaly can be due to different processes such as anticyclonic eddies, the strengthening of a quasi-permanent anticyclonic eddy, the weakening of a cyclonic eddy, or the displacement of a current or eddy (Rio et al. 2007; Pegliasco et al. 2020). Pegliasco et al. (2020) show that it is more appropriate to use absolute dynamic topography rather than SLA to track eddies. For the correct assimilation of altimetry data into models, Hamon et al. (2019) have highlighted the need for an accurate MDT as well as its associated error. Finally, absolute dynamic topography provides access to geostrophic currents, useful data for monitoring ocean currents, and different applications such as maritime security and ocean pollution.

Furthermore, with the advent of new swath observations from the SWOT (Surface Water and Ocean Topography) satellite launched in December 2022 (Fu et al. 2012), which provides sea level observations over swaths 120 km wide with a resolution of 2 km, an accurate MDT at a spatially finer resolution and defined close to the coast is needed.

To deliver the absolute dynamic topography, it is necessary to estimate an accurate Mean Dynamic Topography (MDT; ADT = MDT + SLA) that was removed from the altimeter signal when MSS was subtracted. Since the launch of the ESA GOCE satellite (Gravity and Ocean Circulation Experiment; Pail et al. 2011), the Earth's geoid has been measured with centimetric accuracy at a spatial resolution of 100 km. In addition, the accumulation of altimetry data, improved processing and, in recent years, the special processing applied to leads (fractures in ice) have led to an improvement in the Mean Sea Surface (MSS) and its estimation over ice-covered areas such as the Arctic Ocean (Schaeffer et al. 2023). The "geodetic" approach (Bingham et al. 2008), which consists of estimating the MDT by subtracting the geoid from the MSS, then applying a reliable filter, provides accurate solutions for spatial scales greater than 100 km. To estimate spatial scales shorter than 100 km, it is necessary to add information to these scales. A first method is to use altimetry data to add finer scales to the geoid. These geoids are called combined geoids, Eigen6c4 (Förste et al. 2014) and XGM2019e (Zingerle et al. 2020) are examples. From these combined geoids, it is possible to estimate a geodetic MDT such as MDT DTU22 (Knudsen et al. 2022). Another approach is to use a large-scale satellite-only geodetic MDT and add the small scales from in-situ ocean data (temperature and salinity profiles, velocities estimated from drifting buoys or surface current measuring radars). These in situ data need to be processed to extract only the physical content corresponding to the MDT. This is the approach used in this study and various prior CNES-CLS MDTs (Rio and Hernandez 2004; Rio et al. 2011; Rio et al. 2014; Mulet et al. 2021).

This paper presents the new CNES-CLS22 Mean Dynamic Topography (MDT) solution. Improvement has been made possible by the recent availability of updated time series of drifter and in situ temperature and salinity profiles, and improved MSS and geoid. The method is restated in section 2, while data used in the computation and validation are presented in section 3. The new CNES-CLS22 MDT is described and validated in section 4. Conclusions and discussion are provided in section 5.



## 2 Data

The CNES-CLS22 MDT is calculated from a combination of altimeter and satellite gravity data, in situ measurements, and model winds. The method allows us to estimate the mean over the 1993-2012 reference period but is not limited to observations from this period. For each in situ observation, we remove the altimetric variability referenced to the 1993-2012, thus obtaining an estimate of the mean dynamic topography corresponding to the reference period. The following datasets are used:

– *MSS.* The CNES-CLS22 MSS derived for the 1993– 2012 reference time period by Schaeffer et al. (2023) is used.

– *Geoid model.* The satellite-only geoid model GOCO06s (Kvas et al. 2021) is used with the CNES-CLS22 MSS in the computation of the MDT first guess.

– *Altimeter sea level anomalies (SLAs).* The DUACS-2021 (Taburet et al. 2022) multi-mission gridded sea level and derived geostrophic velocity anomaly products distributed by the Sea Level Thematic Assembly Center (SL-TAC) from the CMEMS altimeter are used.

– *Dynamic heights.* These are calculated from temperature and salinity (T/S) profiles from CTD and ARGO from CMEMS CORA Release November 2022 (period 1993-2020, Szekely et al. 2023), processed by the In Situ Thematic Assembly Center (INS-TAC) of the Copernicus Marine Environment and Monitoring Service (CMEMS)..

– *In situ velocities.* Two types of in situ drifting buoy velocities are used, the 6-hourly SVP-type drifter distributed by the Surface Drifter Data Assembly Center (SD-DAC; Lumpkin and Johnson 2013) and the Argo floats surface velocities from the regularly updated YOMAHA07 dataset for the period 1997–2021 (Lebedev et al. 2007). SVP-type drifters consist of a spherical buoy with a drogue attached in order to minimize the direct wind slippage and follow the ocean currents at a nominal 15 m depth. When the drogue gets lost, the drifter is advected more by the surface currents and also affected by the direct action of the wind. This new CNES-CLS22 MDT also uses velocity data estimated from HF radars located in the Mid Atlantic Bight area, East coast of the US from Cape Hatteras to Cape Cod (processed by Rutgers University, Roarty et al. 2020).

– *Wind data.* Wind stress data are needed for the calculation of the wind-driven velocities (Sect. 2.3) that is used to remove part of the ageostrophic component from drifter velocities. We use the 3-hourly, 80 km resolution wind stress fields from ERA5 (Hersbach et al. 2018) for the period 1993–2021.

## 3 Method

The method used to estimate the new CNES-CLS2022 MDT follows the same approach  detailed by Rio and Hernandez (2004), Rio et al. (2007, 2011 and 2014a), and Mulet et al. (2021). It is a three-step approach summarized below:

The first step is to compute a first guess MDT from the filtered difference between the MSS and the geoid model: a geodetic MDT. The effective resolution of this field depends on the noise level of the raw differences between the MSS and the geoid height; it is around 125 km (Bruinsma et al. 2014).

The second step is to compute synthetic estimates of the MDT and associated mean geostrophic velocities from in-situ data. The drifter data and High Frequency (HF) radar currents are processed to keep only the geostrophic component. The dynamic



heights estimated from the T/S profiles are processed to add missing components: the barotropic component and the deep
baroclinic component. Temporal variability is removed from the dynamic heights and velocities by subtracting the altimeter
sea level anomalies and the associated geostrophic velocity anomalies, respectively. Since the altimeter sea level anomalies
are referenced to the period 1993-2012 , the processed in-situ dynamic heights and velocities are also referenced to the same
interval, and this allows the use of in-situ observations over a longer period than the reference period (Rio and Hernandez
2004). The processed dynamic heights are averaged by 1/4° boxes to obtain the synthetic mean heights and the processed
velocities are averaged by 1/8° boxes to obtain the synthetic mean velocities. Velocities from HF radar are averaged per cell
(6 km by 6 km resolution). Note that this version of the MDT uses only Mid Atlantic Bight HF radar data.

Finally, the third step consists in improving the large-scale MDT (from step 1) with the synthetic data (from step 2) through a
multivariate objective analysis whose formulation was first introduced in oceanography by Bretherton et al. (1976). This
analysis takes as input the a priori knowledge of the MDT variance (as explained in Rio et al., 2011) and zonal and meridional
correlation scales (correlation function proposed by Arhan and Colin De Verdière 1985).

## 3.1 Computation of first guess and comparison with previous first guess

The raw difference between the CNES-CL22 MSS and GOCO06s geoid height is filtered using the optimal filter described by
Rio et al. (2011). For the MDT CNES-CLS22 computation, this step has been improved with the application of an additional
Lagrangian filter along the coast to minimize streamlines of the geostrophic surface current going into land.

The geostrophic velocities associated with the first guess calculated from the raw differences between the CNES-CL22 MSS
and GOCO06s geoid height optimally filtered are compared with the drifter velocities (section 3); the drifter velocities have
been processed to obtain a physical content comparable with the geostrophic velocities. Similarly, the geostrophic velocities
associated with the first guess of the CNES-CLS18 MDT have been compared with the drifter velocities. Figure 1 shows the
improvement (blue color) or degradation (red color) of the RMS of the differences with the drifter velocities of the CNES-
CLS22 first guess compared with the CNES-CLS18 first guess, in current amplitude (a) and current direction (b). Figure 1a
shows that current amplitude is strongly enhanced near the coast, almost everywhere. In the open ocean, the differences
between the two first guesses in comparison to drifters are minimal, except south of 45°S, particularly in the Indian Ocean,
where there are degradation boxes for first guess 2022. For this comparison, Lagrangian filtering at the coast has not yet been
applied. This improvement at the coast in the amplitude of the geostrophic currents associated with the first guess compared
with the drifters is linked to the use of the new CNES-CLS22 MSS and the new GOCO06s geoid. As for the current direction
shown in Figure 1b, there is no clear improvement or deterioration in the new first guess compared with the old one.





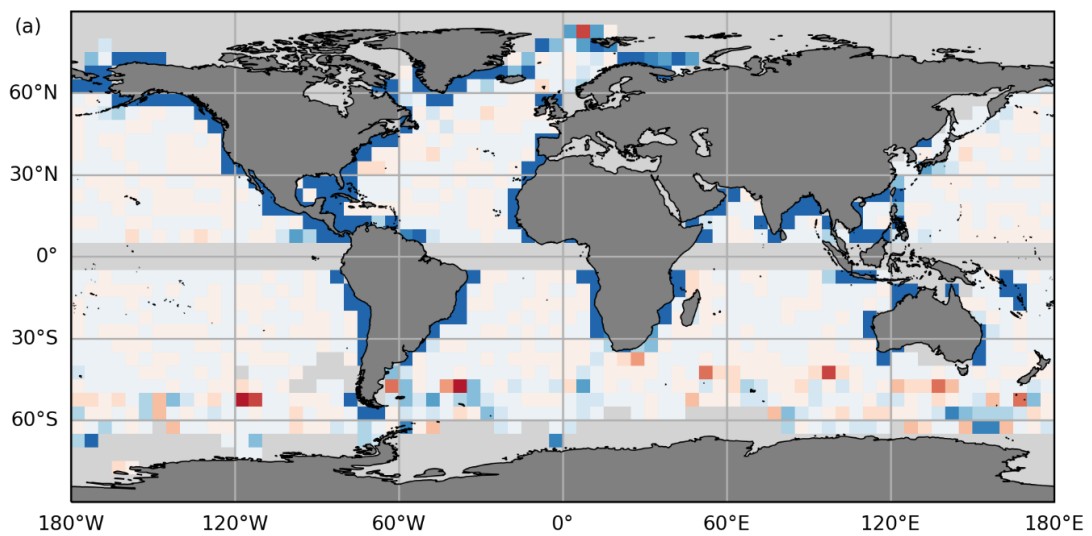

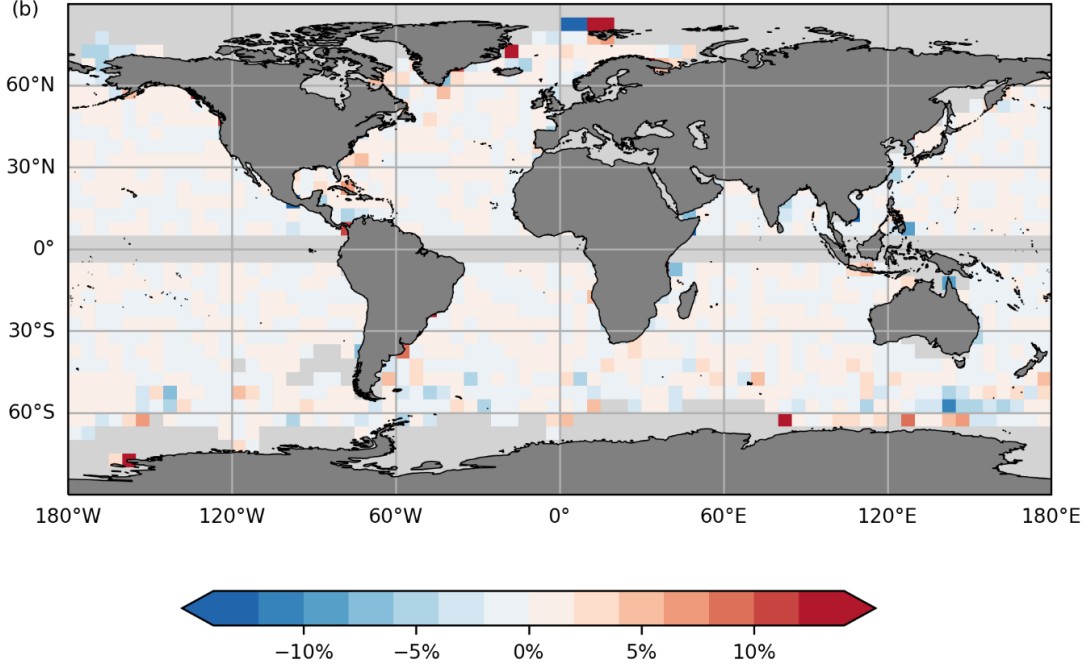

Figure 1: Comparison of the RMS of the differences (a) in current modulus and (b) in direction, of the independent drifting buoy velocities and the altimeter geostrophic velocities obtained using MDT solutions first guess of the CNES-CLS22 (unfiltered at the coast with the Lagrangian filter) and the first guess of the CNES-CLS18, in percent :

$\%RMSD = RMS(U_{MDT_{FG22}}) - RMS(U_{MDT_{FG18}} - U_d)/RMS(U_{MDT_{FG18}} - U_d)$ . These statistics are calculated in boxes of 5° by 5°. Only boxes with more than 100 measurement points and more than 10 different drifters are shown. The bluish colors denote improvement while reddish colors stand for degradation.



### 3.2 Computation of the synthetic mean heights

The estimation of synthetic mean heights aims to reproduce the physical content of the Mean Dynamic Topography (MDT) using temperature and salinity (T/S) profiles, the method is described by Rio et al. (2011) and also used by Rio et al. (2014a) and by Mulet et al. (2021). Dynamic heights are computed relative to various reference depths (200, 400, 900, 1200, and 1900

m), capturing the baroclinic component of ocean circulation. These estimates lack contributions from deeper baroclinic and barotropic processes, the method enables the estimation of the missing components. The optimal analysis is then performed with anomalies relative to the first MDT guess, which is later restored to obtain the final MDT. This approach isolates the small-scale structures of synthetic dynamic heights. The resulting synthetic mean height anomalies reveal strong signals in major ocean currents, such as western boundary currents and the Antarctic Circumpolar Current, where anomalies exceed ±10

cm. These anomalies enhance the geostrophic slopes, thereby accelerating large-scale oceanic currents.

### 3.3 Computation of the synthetic mean velocities

Velocities measured from drifters (section 3) and surface Argo float drifts are processed to obtain estimates of the geostrophic current associated with the MDT. This is achieved by removing from the drifter velocities the ageostrophic components of the

current, as well as the temporal variability of the geostrophic component of the velocities:

$$U_{synth} = U_{drifter} - U_{Ekman} - U_{Stokes} - U_{inertial} - U_{tidal} - U_{ageo-hf} - U_{slippage} - U'_{alti}$$

First, wind-driven currents ($U_{Ekman}$) are removed from the drifter velocity, as well as the wind slippage ($U_{slippage}$), which is the direct effect of wind on undrogued drifters. $U_{Ekman}$ is taken from the Copernicus-Globcurrent product (MULTIOBS_GLO_PHY_REP_015_004,

https://data.marine.copernicus.eu/product/MULTIOBS_GLO_PHY_REP_015_004/description) while wind slippage correction ($U_{slippage}$) is available in the CMEMS INSITU_GLO_PHY_UV_DISCRETE_MY_013_044 product (https://data.marine.copernicus.eu/product/INSITU_GLO_PHY_UV_DISCRETE_MY_013_044/description). These products are consistent as they use same inputs for computation: ERA5 wind and wind stress (section 3) and Mixed Layer Depth as a proxy to the Ekman layer thickness (from CMEMS ARMOR3D:

MULTIOBS_GLO_PHY_TSUV_3D_MYNRT_015_012,
https://data.marine.copernicus.eu/product/MULTIOBS_GLO_PHY_TSUV_3D_MYNRT_015_012/description). Then the temporal variability of the geostrophic component of velocities ($U'_{alti}$) is removed. Next, the tide, inertial oscillations and residual ageostrophic signal (as $U_{stokes}$) are removed by high-frequency filtering. Finally, the mean synthetic velocities obtained are averaged in boxes 1/8° by 1/8°.

This method, the estimation of the wind-driven component of the current and slippage, and the filtering applied are fully described by Mulet et al. (2021, section 5) and builds on previous work by Rio and Hernandez (2004), Rio et al. (2007, 2011 and 2014a).



For this new MDT, we are also using surface velocity data from High Frequency radars located in the Mid Atlantic Bight
region of the East Coast of the USA, from Cape Hatteras to Cape Cod. In the same way as for drifter data, these velocities are
processed to extract the information they hold on the mean geostrophic velocity associated with MDT. We used the cleaned,
detided, high-frequency filtered mean currents for the period 2006-2016 processed by Rutgers University (Roarty et al. 2020).
The mean wind-driven currents over the same period ($U_{Ekman}$ taken from the Copernicus-Globcurrent product
MULTIOBS_GLO_PHY_REP_015_004) were removed and finally these mean currents were re-referenced to the 1993-2012
period.

Figure 2 shows these mean synthetic velocities estimated from (a) drifters (at 1/8° resolution) and (b) HF radar, over the Mid
Atlantic Bight area off New Jersey and Delaware (USA). The figure also shows the 100 m and 2000 m isobaths that define the
extent of the continental shelf. The average velocities estimated by drifters are noisier and more intense than those estimated
from HF radars. This intensity is consistent with the tendency of drifters to accumulate in the shelf-break where currents are
strong. Both maps show recirculation to the south-east at the 100 m isobath. The drifters (Figure 2a) show this current as
narrow and intense (15 to 20 cm/s), whereas the HF radars (Figure 2b) show a broad current of between 5 and 10 cm/s. Very
close to the coast, velocities are generally low (below 5 cm/s), but currents can be perpendicular to the coast (e.g. Figure 2b at
74.5°, 75° and 75.5°W). Outflow from the Delaware Estuary could explain cross-shore currents near 38.5°N, though this is
also near the baseline of the HF radar sites where directional accuracy is diminished



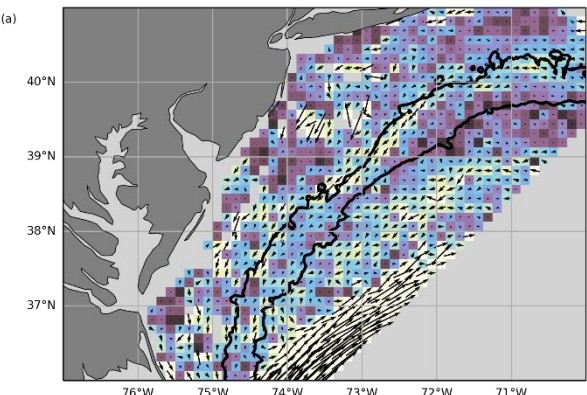

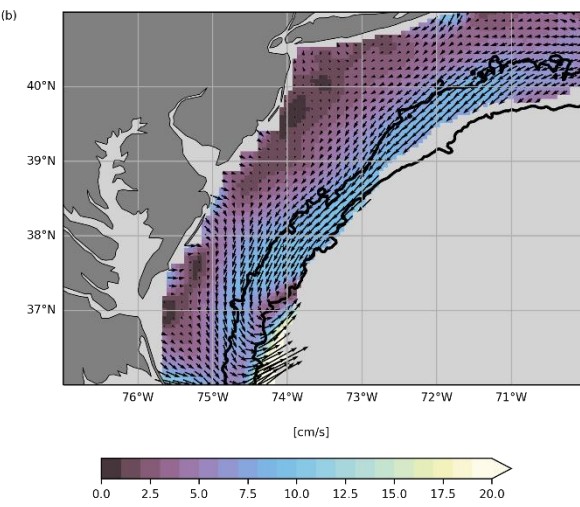


**Figure 2: Synthetic mean velocities in the Mid Atlantic Bight region off New Jersey and Delaware, estimated (a) from drifting buoy (at 1/8°) and (b) from HF radar data. Black lines represent 100m and 2000m isobaths.**

### 3.4 Multivariate objective analysis

The third step of this method is multivariate objective analysis as described by Rio and Hernandez (2004), Rio et al. (2007,

2011 and 2014a) and by Mulet et al. (2021), which uses the synthetic mean geostrophic velocities and synthetic mean heights to improve the first guess, in particular to improve the fine scales, to obtain the CNES-CLS22 MDT. This optimal analysis requires the a priori MDT variance and the a priori zonal and meridional spatial correlation scales of the estimated field. The same statistical a priori as for Rio et al. (2014a) and Mulet et al. (2021) are used here. In the equatorial bandwhere the geostrophic approximation is no longer valid, only mean synthetic height observations are used for MDT estimation, and  only

mean synthetic velocities observations are used for current inversion.



## 4 Results

### 4.1 High-resolution CNES-CLS2022 MDT and associated currents

The CNES-CLS22 MDT obtained is shown in Figure 3a and the magnitude of the associated geostrophic currents is displayed in Figure 3b. Compared with the first guess, the CNES-CLS22 MDT contains more small scales, gradients are sharper, and

currents are accelerated. Figure 4 also shows a zoom of this new CNES-CLS22 MDT (a) and the geostrophic current magnitude (b) over the Arctic zone, as well as a zoom of the CNES-CLS18 MDT (c) and currents (d). Firstly, we note that the CNES-CLS22 MDT covers the Arctic zone, which was not the case before. This is due to the improved coverage of the CNES-CLS22 MSS used to estimate the first guess of this new MDT. Artifacts present on the CNES-CLS18 MDT have disappeared from the new version, for example around 110-120°E. Moreover, Beaufort gyre is better resolved, and Pan-Arctic transport is visible.

On the other hand, the Beaufort Gyre tends to "spread out" over the Canadian Archipelago, which is not physical. In this area of the Canadian Archipelago, the CNES-CLS22 MSS is slightly weaker (Schaeffer et al. 2023), which probably explains the poor physical representation of the CNES-CLS22 MDT. Furthermore, the reliability of MDT is lower in areas where observations are rare or absent, so results must be interpreted with caution in these areas; in the European Arctic, the Kara Sea is an example of such sparsely observed areas.




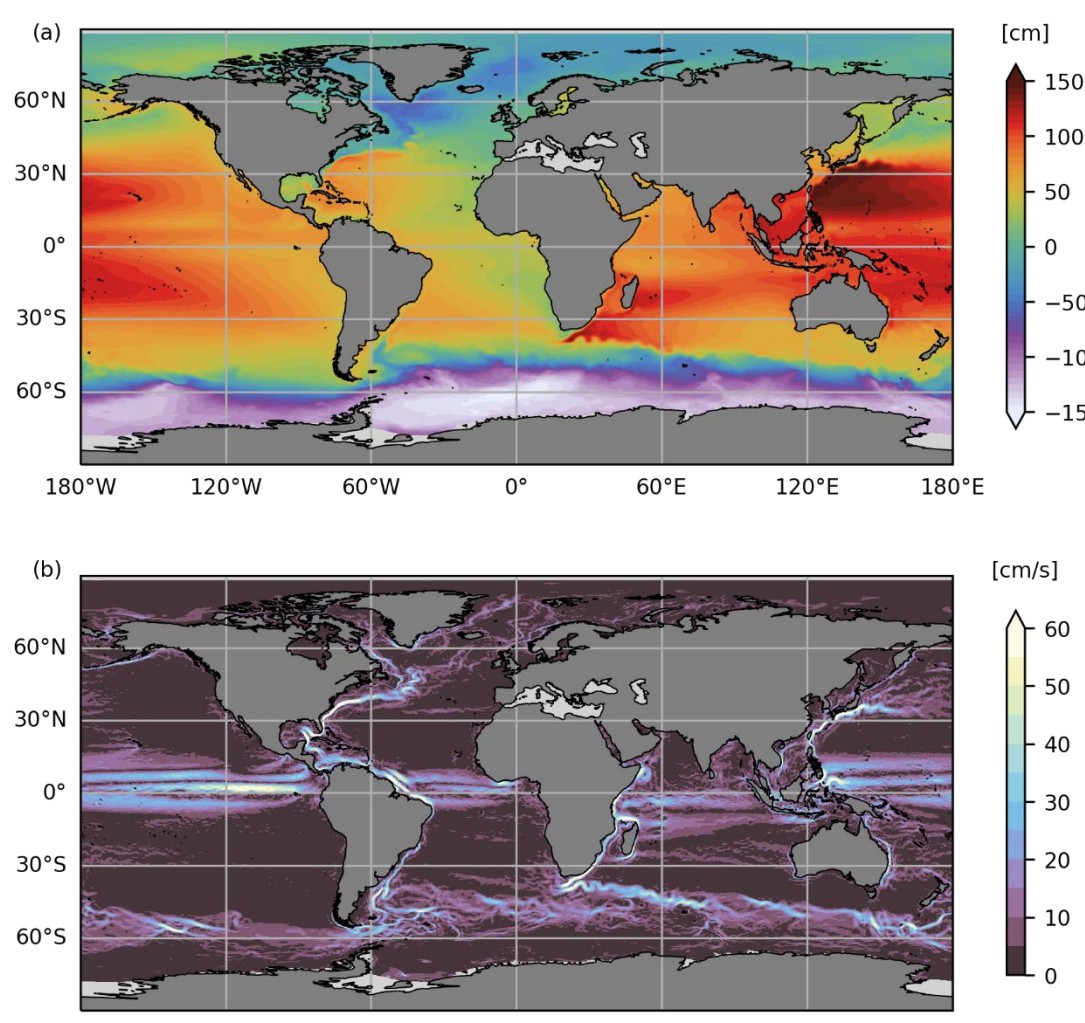

Figure 3: (a) The CNES-CLS22 MDT (cm) and (b) and the amplitude of the geostrophic currents associated with this MDT (cm/s).

**Figure 4: Zoom in on the Arctic zone of (a) the CNES-CLS22 MDT (in cm), (b) the amplitude of the associated geostrophic currents (in cm/s), (c) the CNES-CLS18 MDT (in cm) and (d) the amplitude of the geostrophic currents associated with the CNES-CLS18 MDT (in cm/s).**

To look at the energy content of geostrophic currents associated with MDT, different MDT solutions were compared through a spectral analysis: the first guess (called FirstGuess22), the previous CNES-CLS13 MDT, the previous CNES-CLS18 MDT (Mulet et al. 2021), the DTUUH22 MDT (Knudsen et al. 2022), the new CNES-CLS22 MDT and the Glorys12 numerical model MDT (1/12° numerical model from Mercator-Ocean and distributed within CMEMS: product GLOBAL_REANALYSIS_PHY_001_030,



https://resources.marine.copernicus.eu/?option=com_csw&view=details&product_id=GLOBAL_REANALYSIS_PHY_001_030).

Figure 5 shows spectral analysis for the westward component of mean geostrophic current (noted MDU) associated with MDTs, for a 10° box in the Antarctic Circumpolar Current area, between 38°E and 48°E and between 44°S and 54°S. In this zone, the CNES-CLS22 MDU (in black) has significantly more variance than the first guess (noted FirstGuess22, in dashed blue) on scales from 600 km to the smallest, and it is the in-situ data that increases the variance on these scales. The three CNES-CLS MDTs (13 in yellow, 18 in red and 22 in black) follow the variance level and spectral slope of the GLORYS12

model (in green), up to about 100 km for the CNES-CLS13 MDT, and up to around 60 km for the CNES-CLS18 and the CNES-CLS22 MDT. At the smallest scales, there is still a variance decay with the same slope toward the smallest scales. We might have expected a sharp drop at small scales, as with DTUUH22 MDT which drops off at 40 km, because at less than the Rossby radius we can't infer geostrophic velocity from MDT, so it should become incoherent and lose energy. It is possible that noise in CNES-CLS22 MDT generates this excess variance on small scales.

Figure 5 also shows the spectrum of the DTUUH22 MDT (in dashed pink), whose energy curve decays more rapidly than GLORYS12 below 300 km scales, reaching a plateau around 100 km until 50 km.

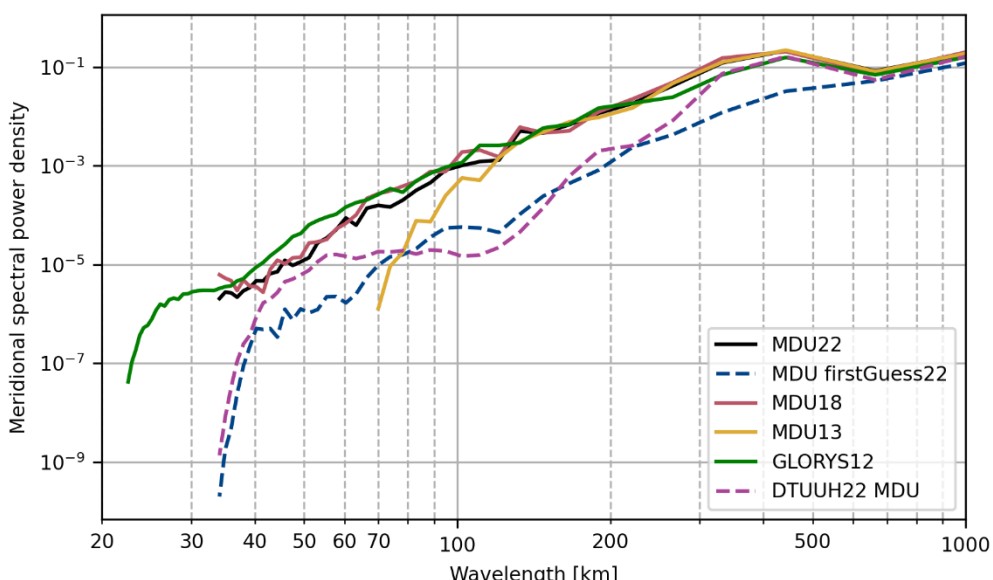

**Figure 5: Spectral Power Density calculated on westward component of geostrophic current associated with different MDT solutions in a 10°x10° box in the Antarctic Circumpolar Current area for the meridional direction, between 38°E and 48°E and between 44°S and 54°S.**



## 4.2 Validation

The validation of the CNES-CLS22 MDT is carried out using different approaches. First, this solution is evaluated qualitatively by region (the European Arctic and the Mid Atlantic Bight), then it is evaluated quantitatively with independent drifter data
and then with independent height data estimated from T/S profiles.

### 4.2.1 Qualitative validation

#### 4.2.1.1 The European Arctic

As seen previously, the CNES-CLS22 MDT provides better coverage of the Arctic region and corrects various CNES-CLS18 artifacts. In this section, we take a closer look at the European Arctic region, and in particular the Yermak Plateau area where
the Fram Strait branch of the Atlantic Water flow to the Arctic enters the Polar Basin (fig. 7a), and the St. Anna Trough in the northern Kara Sea which is the main gateway for the Barents Sea branch of the Atlantic Water flow to the Polar Basin (e.g., Schauer et al., 2002, Rudels, 2015; Fig. 7d). We are looking at different solutions for these two areas. For the zoom on the Yermak Plateau, the new CNES-CLS22 MDT solution is shown in Fig. 7a with the bathymetric and geographic elements cited in this section, the CNES-CLS18 MDT solution is shown in Fig. 7b and in 7c the DTUUH2022 solution is shown. For the
zoom on the St Anna Through, the CNES-CLS22 MDT solution is shown in Fig. 7d with the geographical elements mentioned, the CNES-CLS18 solution is shown in Fig. 7e and the DTUUH2022 solution in Fig. 7f. The first observation is that the DTUUH2022 solution is smoother than the two CNES-CLS solutions on these two zones.

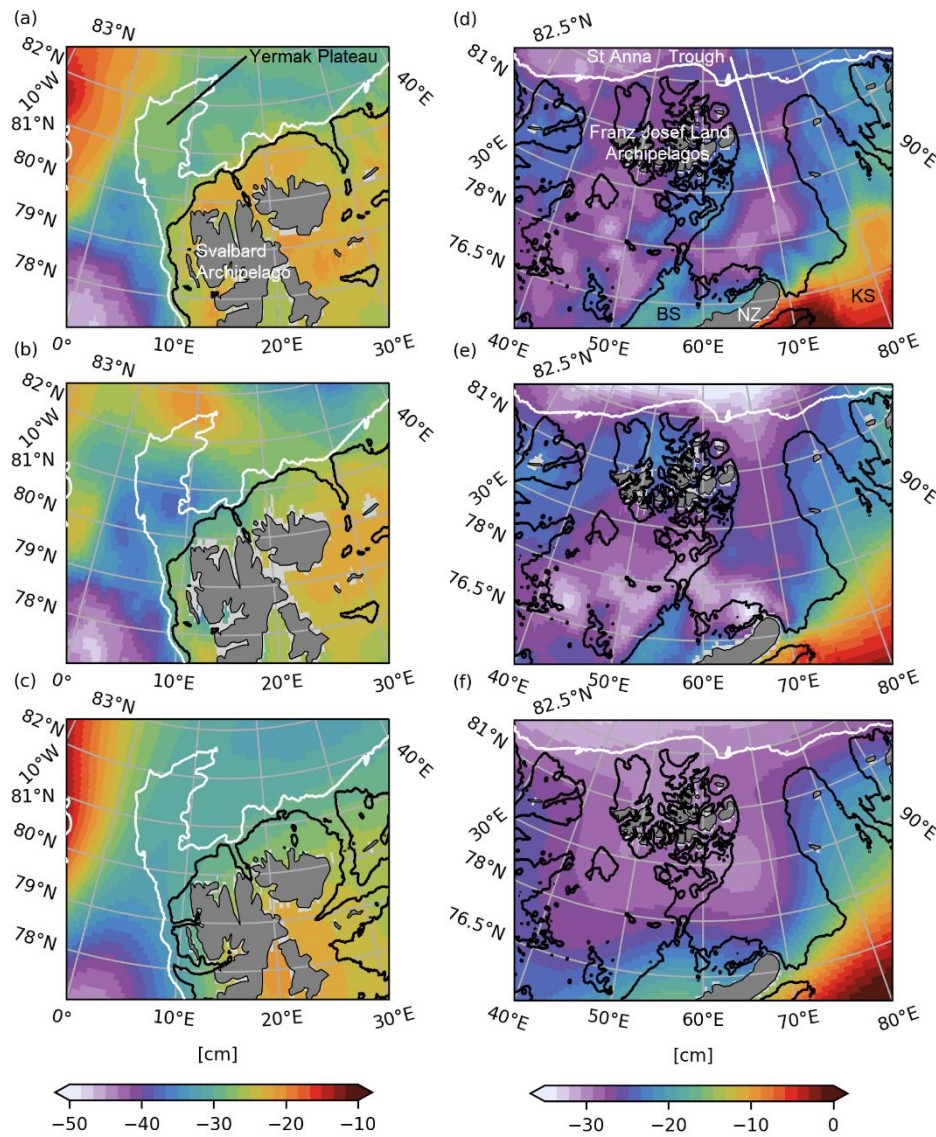

**Figure 6: (a) CNES-CLS22 MDT, (b) CNES-CLS18 MDT and (c) DTUUH22 MDT in Yermak Plateau area. The white line**
**represents the 1000 m isobath and the black line the 200 m isobath. Figure 7a shows the bathymetric and geographical features: the**
**Yermak Plateau and Svalbard Archipelago. (d) CNES-CLS22 MDT, (e) CNES-CLS18 MDT and (f) DTUUH22 MDT in St Anna**
**Through area. The white line represents the 1000 m isobath and the black line the 200 m isobath. Figure 7d shows the geographical**
**features: the St Anna Through, the Franz Josef Land Archipelago, the Novaya Zemlya (NZ), the Barents Sea (BS) and the Kara Sea**
**(KS).**

In the Yermak Plateau area, where the Fram Strait branch of the Atlantic Water inflow enters the Polar Basin, the CNES-
CLS22 MDT shows better alignment of the flow with the bathymetry both for the Svalbard branch crossing the plateau and



the Yermak branch going around the plateau (see Fig. 1 in Meyer et al. 2017), compared with the CNES-CLS18 MDT and DTUUH22 MDT.

The CNES-CLS22 MDT also shows some improvements in the St. Anna Trough where the Barents Sea branch of the Atlantic
Water flow to the Arctic exits the Barents Sea and enters the Polar Basin. Here, the CNES-CLS22 MDT better aligns with the bathymetry in the northeastern Barents Sea and northern Kara Sea compared with the CNES-CLS18 MDT and DTUUH22 MDT. Moreover, several more detailed differences also better align with features of the flow reported in the literature. The main flow from the Barents Sea is clearly aligned along the bathymetry in the southern part of the trough between the Novaya Zemlya and Franz Josef Land archipelagos (Schauer et al. 2002; Lien and Trofimov 2013), and there is a clear indication of
cyclonic circulation of the Fram Strait branch within the same trough (Gammelsrød et al. 2009; Lien and Trofimov 2013). There is also an indication of a continuous Coastal Current along the northern tip of Novaya Zemlya (Lien and Trofimov 2013), which is not clearly seen in the CNES-CLS18 MDT and DTUUH22 MDT. Further north in the St. Anna Trough the CNES-CLS22 MDT shows better alignment of the flow with the bathymetry (e.g., Dmitrenko et al. 2015) compared with the smoother DTUUH22 MDT, in addition to more distinct features not seen in the CNES-CLS18 MDT or DTUUH22 MDT. At
81N, there is an indication of a cyclonic eddy previously identified by hydrographic observations, likely caused by the Fram Strait branch entering the St. Anna Trough (Osadchiev et al. 2022). Moreover, a bathymetric feature to the east of the Franz Josef Land archipelago likely affects the southward flowing part of the Fram Strait branch and causes an anti-cyclonic flow as shown in the CNES-CLS22 MDT.

### 4.2.1.2 Mid Atlantic Bight

As seen in section 2.3, the CNES-CLS22 MDT integrates synthetic geostrophic velocities estimated from high-frequency radar velocities in the Mid Atlantic Bight (MAB) area on the east coast of the USA between Cape Cod and Cape Hatteras. In this section, the CNES-CLS22 MDT is looked at more precisely in the MAB and the adjoining region just north of the Gulf Stream where southwestward flow in the MAB coastal ocean crosses isobaths to depart the continental shelf and join a  recirculation in the Slope Sea (Figure 7b).

Figure 7 shows the contours of the CNES-CLS22 MDT (d) in comparison with the CNES-CLS18 MDT (c) and a ROMS model MDT (a). The ROMS model MDT (Figure 7a) is an average from the ROMS model used in a climatological diagnostic configuration to calculate a kinematically (coastline and bathymetry) and dynamically (ROMS nonlinear model physics) consistent circulation constrained by mean observations of the ocean state and forced by mean surface fluxes and river inflows (details given in Wilkin et al. 2022).

Lentz (2008) and Zhang et al. (2011) have advanced arguments as to the magnitude of the along-shelf sea level gradient in the MAB necessary to complete a momentum balance. We expect gentle southwestward flow throughout the Mid Atlantic Bight, and certain recirculation features in the Gulf of Maine, and encircling Georges Bank that are clearly evident in the ROMS MDT (Figure 7a). The across-shelf sea level gradient is consistent with observed southwestward mean currents. Furthermore, the known pattern of geostrophic coastal currents requires that the MDT contours be largely parallel to the coast, which is the
case with the ROMS MDT and not always the case with the CNES-CLS18 (Figure 7c) and 22 (Figure 7d)  MDTs.

Figure 7: (a) ROMS model MDT contours, (b) Mid Atlantic Bight circulation, (c) CNES-CLS18 MDT contours and (d) CNES-CLS22 MDT contours.

CNES-CLS18 MDT (Figure 7c) shows a slightly more organized circulation on the shelf, although contours on the inner shelf are noisy. Coastal currents in the Gulf of Maine and Scotian Shelf emerge, t but are weak. In addition, there are still MDT contours strongly intersecting the coast, which implies an unphysical geostrophic surface current normal to the coast and sea-level slope inversions along the shelf when sea level at the coast is expected to decrease monotonically toward the south on the basis of along-shelf momentum balance arguments.

The CNES-CLS22 MDT (Figure 7d) remains noisy on the continental shelf, and there are still contours cutting the coast (associated with low geostrophic velocities). Coastal currents on the Scotian Shelf are more organized, but there are still flows

into the coast of central New Jersey. So CNES-CLS22 MDT is a significant improvement, but there is still a way to go to bring MDT to the coast on broad shelves.

### 4.2.1.3 Impact on the Atlantic Water pathways

Atlantic Water (AW) transported to the Arctic Ocean through the Nordic Seas plays a major role in the global climate system. The heat and salt are transported to the Nordic Seas by the Norwegian Atlantic Current (NwAC), a two-branch current system, of which the eastern branch, the Norwegian Atlantic Slope Current (NwASC) follows the shelf edge as a barotropic slope current, while the western branch, the Norwegian Atlantic Front Current (NwAFC) follows the western rim of the Norwegian Sea as a topographically guided frontal current. Further downstream in the Norwegian Sea, the NwAFC and the NwASC,

respectively, form the western and eastern boundaries of the Lofoten Basin, which is the most eddy active region in the entire Nordic Seas. There is no mean flow into the basin and the heat and salt are transported into the basin interior by the mesoscale eddies (Raj et al., 2016, 2020). The quasi-permanent anticyclonic eddy the 'Lofoten Vortex' situated in its western part is the most distinct feature in the Lofoten Basin (Raj et al., 2015). Further downstream, the NwAFC flows along the Mohn Ridge while the NwASC continues along the continental slope, partly branching into the Barents Sea, and flows northwards as the

West Spitsbergen Current.

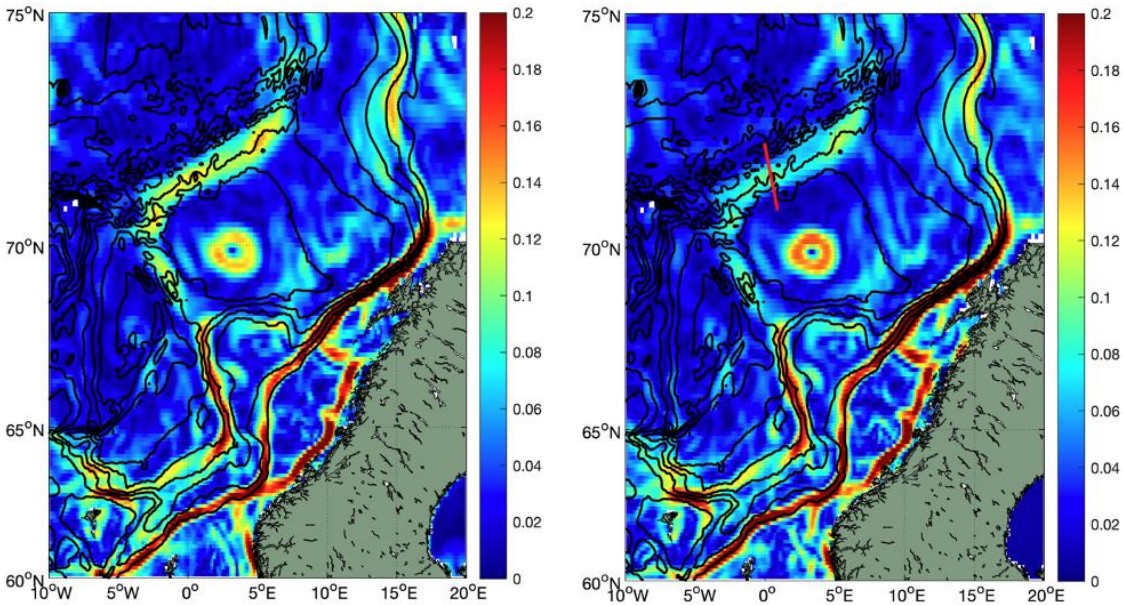

**Figure 8: Mean geostrophic speeds (m/s) derived from: CNES_CLS22 MDT (left panel); and CNES_CLS18 MDT (right). The red line in the right panel indicates the location of the transect shown in Fig. 10. Isobaths are drawn in black.**

The circulation of the Nordic Seas has been the subject of investigations since the Norwegian North-Atlantic Expedition in 1876-1878 (Mohn 1877; Helland-Hansen and Nansen 1909). Monitoring of AW heat transport in the Nordic Seas is mainly performed using numerical ocean model data (Copernicus Arctic Marine forecasting Services; ARC-MFC) and current meters



located at the Iceland Faroe Ridge, the Faroe Shetland Channel, the Svinøy section, and in the Fram Strait. In comparison, Earth observations (EO) from satellites are under-exploited, even though satellite radar altimeters have provided continuous

spatial coverage over the region for more than 30 years. The launch of the Gravity field and steady state Ocean Circulation Explorer (GOCE) mission in 2009, together with the implementation of better retrieval algorithms for the satellite altimeter data processing, has improved our ability to better monitor the ocean circulation (Johannessen et al. 2014). Based on these improvements, Raj et al. (2018) demonstrated the significant value of satellite derived surface velocities for monitoring long term variability of the circulation of AW in the Nordic Seas. Here, we assess the capability of the CNES-CLS22 MDT in better

resolving the circulation of the region.

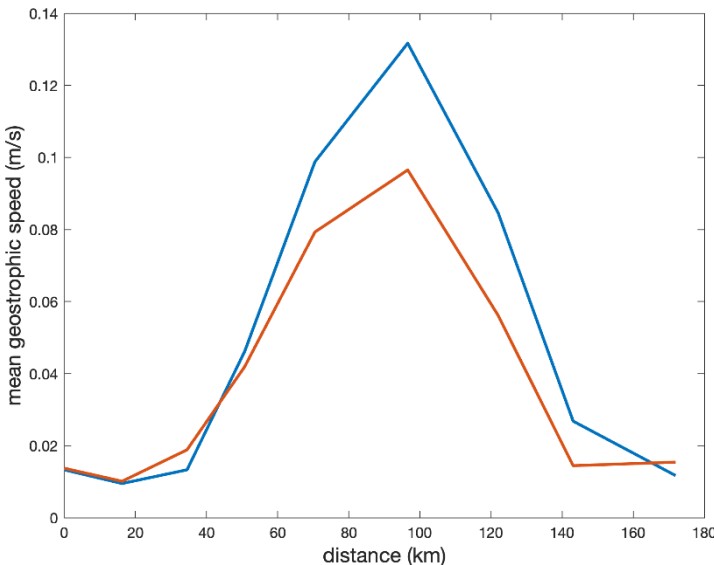

**Figure 9: The mean geostrophic speed derived from CNES-CLS22 MDT (blue) and CNES-CLS18 MDT (red) across the Mohn Ridge (location of the transect shown in Fig. 9).**

Figure 9 shows that both CNES-CLS22 MDT and CNES-CLS18 MDTs reproduce the two-branch structure of the Norwegian

Sea circulation.  The NwASC tightly follows the topography, intensifies at Svinøy and near the eastern border of the Lofoten Basin and due to topographic steering. Along the NwAFC, topographic steering induced intensification of the current is prominent at the western slope of the Vøring Plateau. The main improvement of the CNES_CLS22 MDT derived current estimates are along the western border of the Lofoten Basin and the Mohn Ridge. All previous versions of the MDT produced so far were not able to reproduce the NwAFC in this region correctly. Figure 10 shows that the difference can even reach up

to 4 cm/s. On the other hand, the mean speed of the Lofoten Vortex derived from CNES-CLS22 MDT is comparatively lower than the one derived from the CNES-CLS18 MDT.  All, in all, the results from CNES-CLS22 MDT are promising.



### 4.2.2 Regional quantitative validation

#### 4.2.2.1 Comparison with ADCP in the Faroe-Shetland Channel

The Faroe-Shetland Channel (FSC; Figure 10a), situated between the Shetland shelf to the east and the Faroe plateau to the
west (Chafik 2012), is a crucial passage for the global overturning circulation (Hansen and Østerhus 2000; Chafik et al. 2020).
The region hosts northward flow of warm North Atlantic water carried by the Slope Current toward the Nordic Seas and the
Arctic Ocean (Chafik et al. 2015), and the southward branch of the densest overflow water produced at higher latitudes (Chafik
et al. 2023). The FSC is also partially supplied from north of the Faroes by the inflow of warm waters through the Iceland-
Faroe Ridge (Poulain et al. 1996; Rossby et al. 2018), which recirculates in the channel before merging with the Slope Current
(Poulain et al. 1996; Berx et al. 2013), as illustrated in Fig. 11 (a). Two essential components thus must be included in the
MDT for an accurate representation of the mean circulation in the region: the bathymetrically constrained Slope Current to the
east as well as the strong recirculation that traces the morphology of the FSC (Figure 10a-b).

Figure 10c-d compares the two MDT solutions in the region and validate these against ship-mounted ADCP velocities across
the FSC (Rossby and Flagg 2012; Rossby et al. 2018; data available on: http://po.msrc.sunysb.edu/Norrona/) and, Figure 10e-
f. The CNES-CLS22 MDT shows an improved representation of the large-scale circulation in the eastern subpolar North
Atlantic as well as the FSC (see, e.g., Childers et al. 2015; their Fig. 7). Notably, the spatial structure of the highlighted -0.15
m contour (Figure 10c, blue contour) in the CNES-CLS22 MDT closely follows the bathymetry and passes directly through
the FSC entrance, suggesting a more accurate depiction of the inflow of warm Atlantic waters into the channel (cf. Figure 10a).
In contrast, the -0.15 m contour in the CNES-CLS18 MDT (Figure 10c, red contour) exhibits unrealistic deviations, especially
near the FSC entrance, where it overshoots into the Faroe shelf and misses the channel entrance. Due to this unrealistic
overshoot in CNES-CLS18, there is a pronounced negative difference between the two MDT solutions in the southwestern
part of the FSC, particularly over the Faroe shelf (Figure 10d). However, the CNES-CLS22 MDT introduces a distortion by
deflecting the path of the Slope Current into the channel interior in the northeastern FSC (Figure 10d). This results in a
pronounced positive difference compared to CNES-CLS18 MDT, where the contours are more closely aligned with the
bathymetry, indicating a stronger topographic control of the northward-flowing warm waters.

Because of a more realistic Slope Current in CNES-CLS MDT18, its velocities normal to the FSC are found to closely match
the ADCP estimates as compared to CNES-CLS MDT22 (Figure 10e), where the northward flow is less confined to the slope
and spread out over a larger area across the channel (see also Figure 10c). This behaviour may be due to a restricted and too
narrow recirculation on the Faroe slope, as indicated by the -0.2 m contour (see Figure 10c). To further illustrate this point, we
estimate the cumulative transport across the FSC. For simplicity, we assume a barotropic velocity structure over the upper 400
m layer (see e.g., Rossby and Flagg 2012). Figure 10f shows a good agreement between the CNES-CLS MDT18 net northward

transport of 2.9 Sv and the ADCP-based estimate. In contrast, the CNES-CLS MDT22, which is characterized by a broad or diffuse Slope Current, significantly overestimates the net northward transport at 5.5 Sv.

We conclude that while the CNES-CLS MDT22 offers improvements in representing the large-scale circulation more realistically, this solution comes at the expense of accuracy in regions with narrow currents and strong recirculations. The challenge in accurately constraining the path of the sharp Slope Current may arise from misrepresentations of the recirculating

waters within the FSC. Other contributing factors could include inaccuracies in the representation of the surrounding shelf regions that border the FSC.

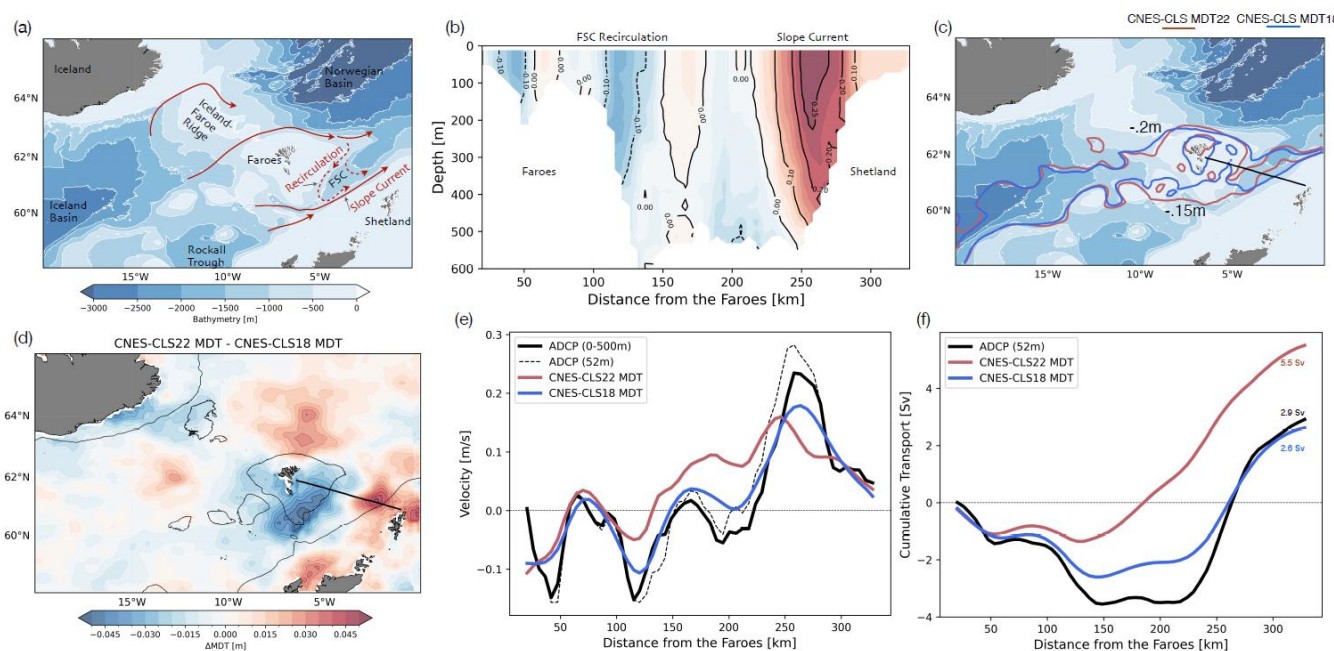

**Figure 10: (a) Bathymetric map including a schematic of upper-ocean circulation in the region. The Slope Current and the**
**recirculation in the Faroe-Shetland Channel (FSC) are indicated. Bathymetric contours are shown at depths of 500, 1000, 1500, 2000, 2500, and 3000 m. (b) Time-mean (2008-2018) velocity structure across the FSC based on ship-mounted Acoustic Doppler Current Profiler (ADCP) between Shetland and the Faroe Islands (see section in panel (c)). The velocities are normal to the ship track. (c) Similar to panel (a), but showing the -0.15 m and -0.2 m contours from two MDT solutions. (d) Regional difference between the two MDT solutions (CNES-CLS MDT22 minus CNES-CLS MDT18). (e) Normal velocities from the ADCP averaged over the**
**upper 500 m (solid black line) and at 52 m depth (dashed black line), compared to CNES-CLS MDT22 (red) and CNES-CLS MDT18 (blue). (f) Cumulative transport integrated from west to east (starting from the Faroes) for the ADCP and both MDT solutions, with transport multiplied by 400 as a rough estimate of upper-ocean transport. For ADCP data, velocities at 52 m are used.**

### 4.2.3 Quantitative validation with independent T/S profiles

Here, CNES-CLS22 and CNES-CLS18 MDTs are compared with independent dynamic height data derived from T/S profile
observations since 2017 that were withheld from the analysis, as noted previously. Dynamic heights estimated from T/S profiles omit the signal due to baroclinic processes occurring below the reference depth, and by barotropic processes,



Therefore, for validation purposes we choose to keep only the deepest profiles (reference depth 1900 m) to minimize the omitted dynamic signal, which leaves us with a validation set of 2% of the database.

As a first step, we compare the CNES-CLS18 and CNES-CLS22 MDTs against these independent dynamic heights by looking at the correlation of the ADT (SLA+MDT considered) and the independent dynamic heights (figures not shown). Correlations are calculated in boxes of 5° by 5° (with at least 20 data) and are high, mostly between 0.8 and 1 for both MDTs, and it is difficult to differentiate between them.

Secondly, Figure 11a shows the mean bias per 5°X5° box between the ADT estimated from the CNES-CLS22 MDT and the independent dynamic heights. This global mean bias is 1.30 m, with spatial variations in the Norwegian Sea and to the south near Antarctica (equivalent for CNES-CLS18, not shown) and mainly represents the barotropic component not observed by the dynamic heights. This is why it is removed from the estimate of mean synthetic heights for the calculation of the MDT and the following validation diagnosis on Figure 11b shows a comparison between the variances of the differences (not considering the bias) between the different ADTs and the dynamic heights, in percent. In blue (in red), the variance of differences is reduced (increased) using CNES-CLS22 compared with CNES-CLS18.

Globally, we see an improvement in CNES-CLS22 MDT compared with CNES-CLS18, but this is not true in all regions. South of the Atlantic and the Indian Ocean (as far south as Australia), the variance of differences is reduced for CNES-CLS22 by more than 10% for many boxes (even if boxes of strong reduction are juxtaposed with boxes of increased variance of differences). Areas of degradation are concentrated in the north-western Atlantic, particularly south of Greenland, in the very north of the Pacific (along the Gulf of Alaska to the Fox Islands, and close to Russia) and in the south of the Pacific, where there are degradations of over 10% (also juxtaposed with improvement boxes).

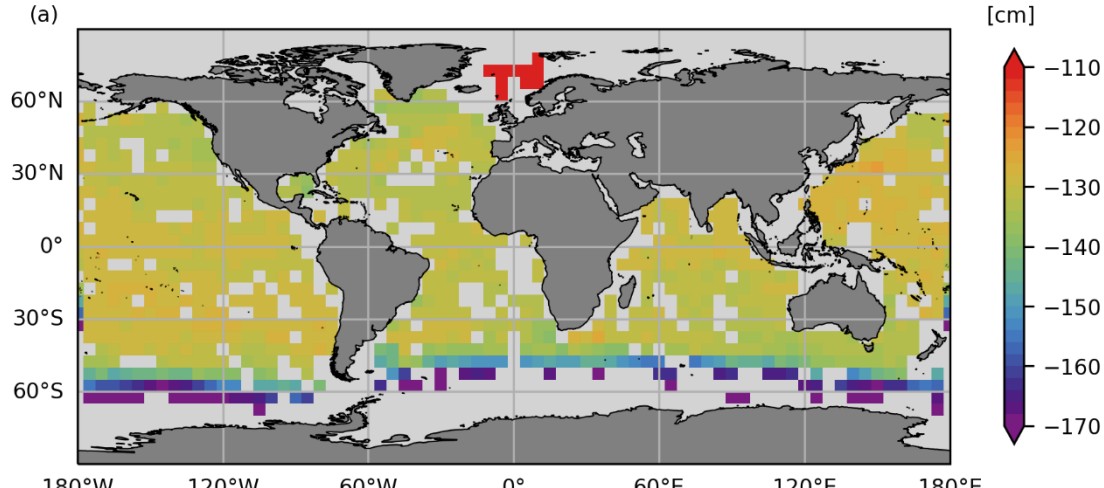

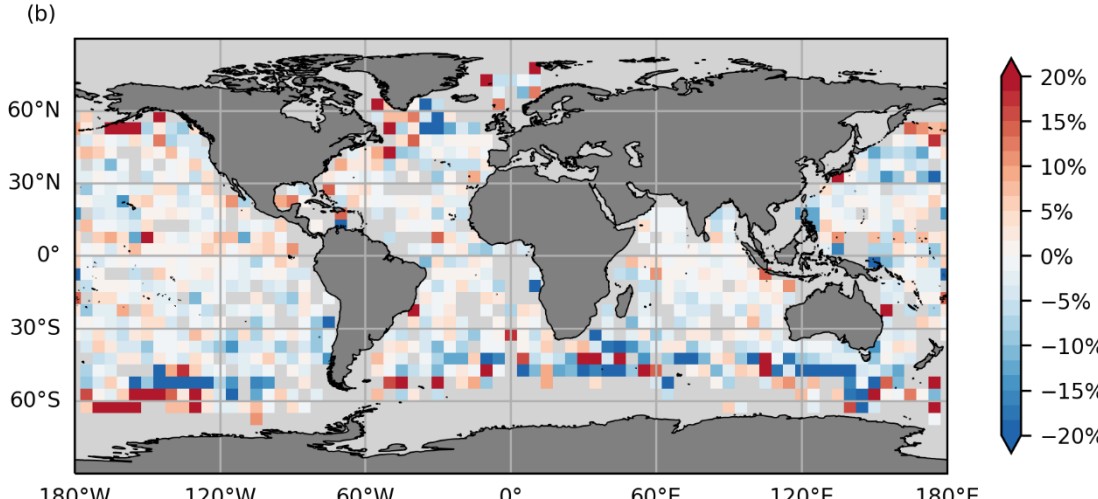

**Figure 11: (a) shows the bias in centimetres between the dynamic heights estimated from the independent T/S profiles and the ADT calculated from the CNES-CLS2 MDT. And (b) shows the reduction/increase in variance of the differences between independent dynamic heights and ADT (from CNES-CLS18 and CNES-CLS22) in percent. In blue (in red), the variance of differences is reduced**

**(increased) using CNES-CLS22 compared with CNES-CLS18. All these statistics are calculated in 5°X5° boxes, and only boxes with at least 20 data are kept.**

### 4.2.4 Quantitative validation with independent drifters

Here, CNES-CLS18 and CNES-CLS22 solutions are compared with independent velocity data. 10% of the AOML drifters

dataset are randomly selected and kept for validation (independent data). These drifters are not evenly distributed across the





oceans. There are few drifters close to Antarctica (south of about 50°S), at the equator, and in Arctic. In addition, the Atlantic is slightly better sampled than the other basins, and the North Indian Ocean less well sampled.

Following the same processing steps described previously, we remove the wind-driven current (Ekman and wind slippage) from total drifters current and then data are filtered at inertia frequency (if inertia frequency is between 1 and 5 days, otherwise

take a minimum of 1 day and a maximum of 5 days) to remove the tide and inertial waves. The objective is to keep only the geostrophic signal. Geostrophy can't be used to estimate currents at the equator, so we exclude drifters between 5°S and 5°N. Absolute dynamic topography values were calculated by adding the CMEMS gridded SLA to the new CNES-CLS22 MDT. Associated geostrophic currents were then derived and interpolated along the drifter trajectories. Bias and Root Mean Square differences (RMSD) between the obtained geostrophic velocities with CNES-CLS22 and CNES-CLS18 and the drifter derived

geostrophic velocities were calculated spatially by 5°X5° boxes. All these statistics are calculated with at least two different drifters and with at least 100 measurement points.

Comparisons of bias (a) in current modulus (in m/s) and (b) in direction (in degrees) are shown in Figure 12. In blue (in red), there is a decrease (increase) in bias for geostrophic currents derived from the ADT estimated with the CNES-CLS22 MDT. Globally, there is a decrease in bias in current modulus (Figure 12a) using the new solution, with a greater decrease in bias in

areas of strong currents: Gulf Stream, Kuroshio, Agulhas Current and Antarctic Circumpolar Current. The areas of degradation are South Greenland and South Kerguelen. For the directional bias (Figure 12b), the contribution of the new solution is more mixed, as the direction of geostrophic currents derived from the ADT using the CNES-CLS18 is better in the South between the Kerguelens and southern Australia. These remain areas with fewer validation drifters.



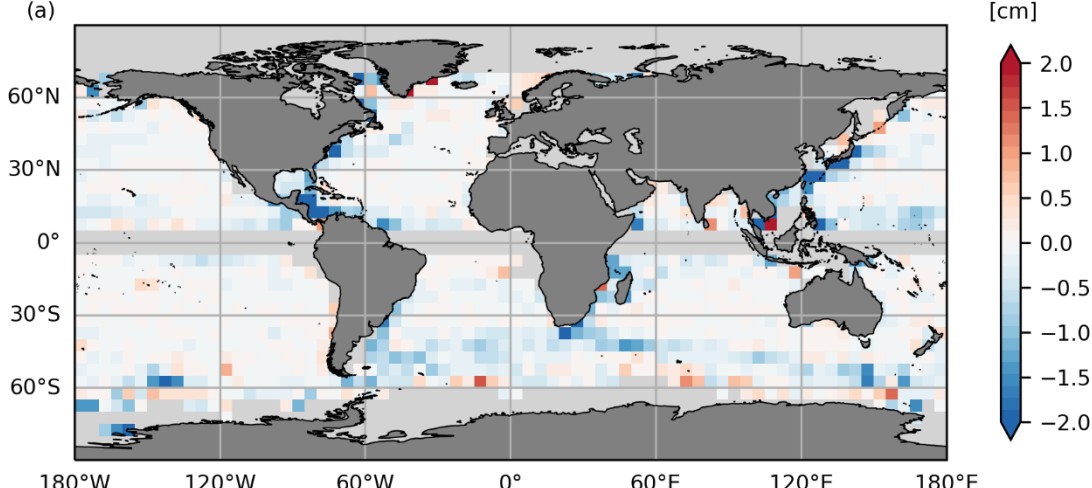

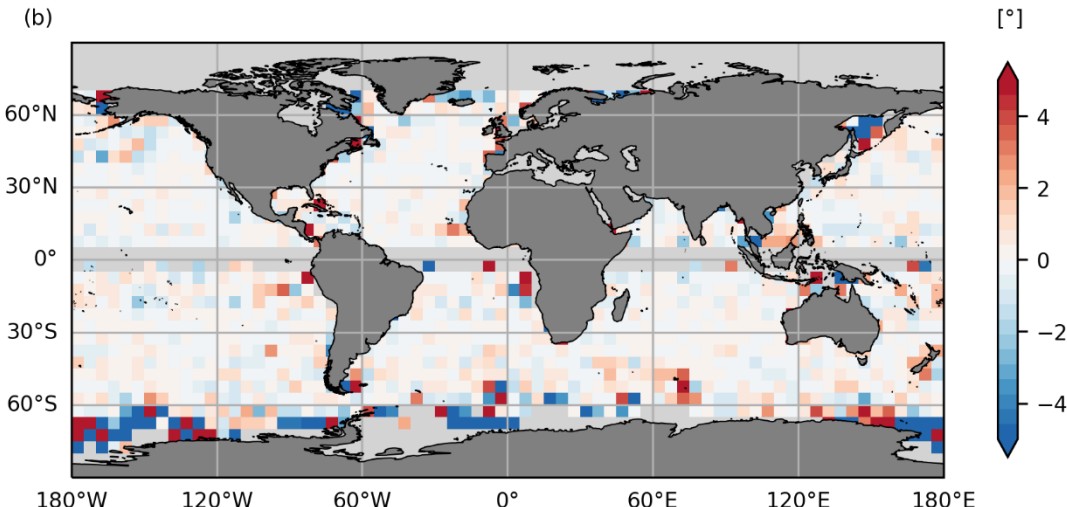

**Figure 12: Comparison of bias (a) in current modulus and (b) in current direction independent drifting buoy velocities and the altimeter geostrophic velocities obtained using different MDT solutions. In blue (in red), the bias is reduced (increased) using CNES-CLS22 compared with CNES-CLS18.**

Earth System Discussions
Science
Data

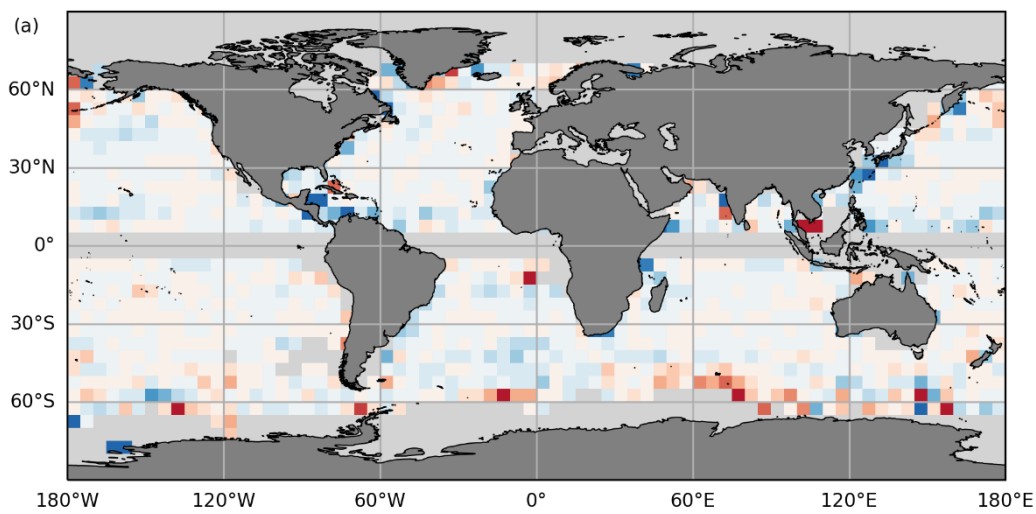

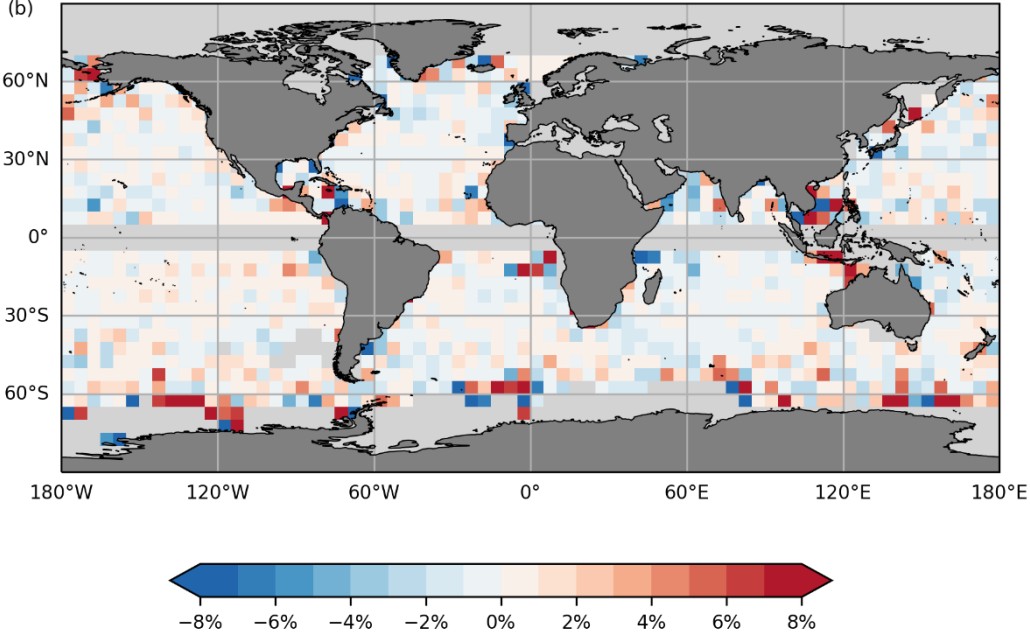

**Figure 13: Comparison of the RMS of the differences (a) in current modulus and (b) in direction, of the independent drifting buoy**
**velocities and the altimeter geostrophic velocities obtained using MDT solutions CNES-CLS22 and CNES-CLS18, in percent :**
$\%RMSD = RMS(U_{ADT22}) - RMS(U_{ADT18} - U_d)/RMS(U_{ADT18} - U_d)$ **. In blue (in red), the bias is reduced (increased) using**
**CNES-CLS22 compared with CNES-CLS18. All these statistics are calculated in 5°X5° boxes, and only boxes with at least 2 drifters**
**difters differents and at least 100 measurement points, are kept.**

Comparisons of RMS of the differences (a) in current modulus and (b) in direction, of the independent drifting buoy velocities
and the altimeter geostrophic velocities obtained using MDT solutions CNES-CLS22 and CNES-CLS18, in percent, are shown
are shown on Figure 13. In blue (in red), the RMSD is reduced (increased) using CNES-CLS22 compared with CNES-CLS18.



In terms of current amplitude (Figure 13a), the majority of boxes show an improvement of between 0 and 2.5% for the CNES-CLS22 solution, with a greater improvement at Kuroshio and in the Caribbean Sea. Areas of degradation in current amplitude RMS are the area around Kerguélen in the ACC and along the southeast coast of Greenland. Comparison of the RMS of

differences in current direction (Figure 13b) shows a more mixed result with degradation boxes compared to the CNES-CLS18 solution: in the ACC in the South Pacific. It can also be noted that some boxes show an improvement in the RMS of differences for the CNES-CLS22 solution in current amplitude but not in direction, and vice versa; this is the case for some boxes south of the Kerguélen Islands and at the extreme south of the Pacific. The areas around the Kerguélen Islands, in the South Pacific between -140°E and -120°E, and around the Fox Islands in the extreme North Pacific remain areas of degradation in RMS of

the differences in modulus and current direction for the new solution compared with the previous one. On the other hand, the Kuroshio, the North Atlantic and the Labrador Sea, as well as the Indian Ocean are areas where the RMS of the differences for the new CNES-CLS22 solution has improved compared to the CNES-CLS18 solution.

Globally, the two solutions CNES-CLS22 and CNES-CLS18 remain close with these drifter comparisons, as shown in the following Table 1, which summarizes the drifter comparison statistics over the whole area up to 70°N. Both solutions have a

difference RMS of between 10.5 and 11 cm/s for U and V components and current modulus, and a difference RMS of around 34° in direction. The comparison of the two solutions favours the new one, but by less than 1%. It should also be noted that the comparison between the first guess and the CNES-CLS22 MDT shows an improvement in U and V of around 5% in the RMS of differences, and an improvement of around 6% in current amplitude and 2% in direction, which is expected.

| RMS(differences) | U [cm/s] | V [cm/s] | M [cm/s] | D [°] |
|---|---|---|---|---|
| **CNES-CLS22 MDT** | 10,57 | 10,62 | 10,79 | 34,24 |
| **CNES-CLS18 MDT** | 10,59 | 10,63 | 10,83 | 34,26 |
| **CNES-CLS22 MDT first guess** | 11,11 | 11,17 | 11,47 | 35,05 |
| | | | | |
| **% 22 vs 18** | -0,2% | -0,1% | -0,4% | -0,1% |
| **% 22 vs 22 first guess** | -4,9% | -4,9% | -5,9% | -2,3% |

**Table 1: RMS of the differences for the zonal component U, the meridional component V, the current modulus and the direction of the current, of the independent drifting buoy velocities and the altimeter geostrophic velocities obtained using MDT solutions CNES-CLS22, CNES-CLS18 and the CNES-CLS22 first guess. The last two lines show the reduction/increase in RMS of the differences between two solutions: CNES-CLS22 versus CNES-CLS18 and CNES-CLS22 versus CNES-CLS22 first guess, in percent (a negative percentage is a decrease in the RMS of the differences). These statistics were generated using all available drifters up to 70°N.**

Given the small number of independent drifters in Arctic, this area is treated separately. Table 2 summarizes the RMS of the differences between the geostrophic currents derived from drifters (degraded treatment) and the geostrophic currents derived from ADTs calculated with the CNES-CLS22, CNES-CLS18 and background CNES-CLS22 solutions. The last two lines show in percent the reduction in RMS of the differences between CNES-CLS22 and CNES-CLS18 MDT and its first guess.



In the zonal and meridional components of the current, the new solution reduces the RMS of the differences compared with the drifters by 19% and 12% respectively. This improvement translates into a clear improvement in direction of 9%, which reduces the RMS of the differences from around 56° to 50°; but the RMS of the differences in current amplitude remains very slightly better for CNEs-CLS18. Note that, as the coverage is not quite identical, there are more points taken into account for CNES-CLS22 than for CNES-CLS18.

| RMS(differences) | U [cm/s] | V [cm/s] | Modulus [cm/s] | Direction [°] |
|---|---|---|---|---|
| **CNES-CLS22 MDT** | 7,95 | 8,12 | 9,61 | 50,43 |
| **CNES-CLS18 MDT** | 9,86 | 9,23 | 9,58 | 55,88 |
| **CNES-CLS22 MDT first guess** | 7,97 | 8,26 | 10,02 | 51,77 |
| | | | | |
| **% 22 vs 18** | -19,4% | -12,0% | 0,4% | -9,8% |
| **% 22 vs 22 first guess** | -0,2% | -1,7% | -4,1% | -2,6% |

Table 2: RMS of the differences for the zonal component U, the meridional component V, the current modulus and the direction of the current, of the independent drifting buoy velocities and the altimeter geostrophic velocities obtained using MDT solutions CNES-CLS22, CNES-CLS18 and the CNES-CLS22 first guess. The last two lines show the reduction/increase in RMS of the differences between two solutions: CNES-CLS22 versus CNES-CLS18 and CNES-CLS22 versus CNES-CLS22 first guess, in percent (a negative percentage is a decrease in the RMS of the differences). These statistics are based on the Arctic zone north of 80°N. As the CNES-CLS18 MDT has a smaller coverage, the number of points used is slightly different for this solution than for the others.

**5 Conclusions**

The main improvement of this new CNES-CLS22 MDT over the previous CNES-CLS18 MDT is in the Arctic, with better coverage and a more physical solution (with the disappearance of artifacts from the previous version). Globally, the new CNES-CLS22 solution is close to the CNES-CLS18 solution, both have better resolution of small scales than previous CNES-CLS MDTs but are potentially polluted by noise at very short scales. At these small scales, the solution may introduce unrealistic mean kinetic energy. The CNES-CLS22 MDT has been evaluated against independent height and velocity data in comparison with the previous version, the CNES-CLS18. The new solution presents slightly better results, although not identical in all regions of the globe. In particular, the results are better in the Antarctic Circumpolar Current in terms of height, better off Japan and particularly in the Arctic in terms of geostrophic velocities. For geostrophic currents, those of the new CNES-CLS22 MDT in the Antarctic Circumpolar Current are slightly worse than those of the CNES-CLS18 in comparison with drifters.

Improvements to this new MDT include a new first-guess with the CNES-CLS22 MSS and the GOCO06s geoid to which optimal filtering has been applied, as well as Lagrangian filtering at the coast to reduce the intensity of normal currents at the coast, drifting buoy and T/S profile databases have been updated, as have updated processing to obtain synthetic mean



geostrophic velocities and synthetic mean heights. In addition, a new type of data, land-based HF radar surface current observations, was processed to extract physical content consistent with MDT in the Mid Atlantic Bight region. The study of this region, in particular, showed the improvements of CNES-CLS22 MDT, but that there is still work to be done to obtain a more physical solution on the continental shelf. Indeed, on broad shallow continental shelves in general, the first guess is not always good because it's close to the coast and today we only process T/S profiles with a depth greater than 200 m; we therefore

lack data on these shelves. HF radar data can provide some velocity information, as there are few drifting buoy data in these coastal regions, but processing is difficult, as complex and non-negligible ageostrophic currents have to be removed. In the case of the Mid Atlantic Bight, these HF radar data do not allow us to obtain a MDT with a geostrophic flow not crossing the coastline.

Thanks to the new first-guess and in particular the new CNES-CLS22 MSS, the Arctic area is covered in this MDT and the
artifacts of CNES-CLS18 have disappeared. Looking more specifically at the European Arctic region, the new CNES-CLS22 solution presents structures in agreement with the literature in the Yermak Plateau area and in the St Anna Through region.

In addition, the CNES-CLS22 MDT has been looked at more closely in the Nordic Seas to look at the transport paths from the Atlantic Water to the Arctic Ocean, and the MDT has the ability to better resolve the region's circulation.

In the Faroe-Shetland Channel, the CNES-CLS MDT22 offers a more realistic representation of the large-scale circulation,
but this solution comes at the expense of accuracy in regions with narrow currents and strong recirculations. The difficulty in accurately constraining the trajectory of the sharp slope current may stem from erroneous representations of recirculating waters in the Faroe-Shetland Channel.

With the arrival of new swath observations from the SWOT (Surface Water and Ocean Topography) satellite launched in December 2022 (Fu et al. 2012), the continuous improvement of MDT accuracy and resolution is necessary. The inclusion of
new coastal data, such as HF radar and SAR data, and shallower T/S profiles not currently taken into account, requires a clear separation of geostrophic and ageostrophic processes, and access to physical content consistent with MDT. These treatments call for new methods in order to best process continental shelf areas with varied oceanographic phenomena.

**Data availability**

CNES-CLS22 MDT described in this manuscript can be accessed at https://www.aviso.altimetry.fr/en/data/products/auxiliary-products/mdt/mdt-global-cnes-cls.html under data DOI (Jousset, S. : Mean Dynamic Topography MDT CNES_CLS 2022 (Version 2022), CNES [Data set], https://doi.org/10.24400/527896/A01-2023.003, 2023).



## Author contribution

SJ and SM processed the data and calculated the CNES-CLS22 MDT. LC calculated mean ADCP currents in the Faroe-Shetland Channel area. JW, LV, LC, RR and AB analyzed and validated the MDT in their area of expertise (i.e. Mid Atlantic Bight, European Arctic, Faroe-Shetland Channel and Nordic Seas, respectively). EG, GD and NP supervised the study. SJ prepared the article with contributions from all co-authors.

## Competing interests

The authors declare that they have no conflict of interest.

### Acknowledgements

We thank the beta users for their valuable feedback, and Laurent Bertino for his comments on the Arctic region.

### Financial support

This study has been funded by the French space agency CNES.

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
