# Peer review of "New Global Mean Dynamic Topography CNES-CLS-22 Combining Drifters, Hydrography Profiles and High Frequency Radar Data"

_Earth System Science Data, 2025_

## Referee Comment (RC1)

Review of "New Global Mean Dynamic Topography CNES-CLS-22 Combining Drifters, Hydrography Profiles and High Frequency Radar Data" (essd-2025-429)

**General Assessment**

This manuscript presents the CNES-CLS22 global Mean Dynamic Topography (MDT), an update of the CLS18 solution, incorporating improved MSS and geoid data, updated in situ datasets, and for the first time, HF radar observations. The paper is well aligned with ESSD's data-descriptor mission: it carefully documents methodology, datasets, and validation results, while highlighting both improvements and remaining challenges.

The new MDT is an important community resource and will be widely used in ocean circulation studies, satellite altimetry applications, and model assimilation. The paper is thorough and well-written overall. However, the improvements relative to CLS18 are modest globally, and the discussion of limitations could be more critical.

I recommend publication after moderate revision.

**Major Comments**

**Magnitude of Improvement**

Global validation statistics (Tables 1–2) show only ~0.2–0.5% improvement relative to CLS18 in drifter-based comparisons. The improvements are regionally significant (Arctic, Nordic Seas) but not globally transformative.

The authors should explicitly acknowledge this limited global gain, while framing CLS22 as a necessary incremental update with specific regional advances.

**Small-Scale Noise**

The spectral analysis (Fig. 5) shows excess variance at scales <60 km, suggesting noise contamination. This is a critical issue because MDT is often used at mesoscale/submesoscale resolutions.

The authors should discuss: (i) the likely sources of this noise (filtering? in situ sampling? MSS errors?), (ii) whether users should apply additional smoothing before analysis, and (iii) the implications for applications such as SWOT.

**Continental Shelf Limitations**

Despite the inclusion of HF radar data, MDT remains unphysical on broad shelves (e.g., Mid-Atlantic Bight), with contours crossing coasts.

The discussion should be more candid: Why is MDT particularly problematic over shelves (barotropic dominance, unresolved tides, estuarine flows)? What realistic directions could improve it (e.g., assimilation frameworks, barotropic tide corrections, inclusion of shallow T/S profiles)?

**Validation Coverage**

Drifters are sparse in the Arctic and Southern Ocean; HF radar data are localized. The T/S profile validation is limited to only  $\sim 2\%$  of the dataset (deepest profiles).

Please highlight these gaps more strongly and caution users about interpreting CLS22 in undersampled regions.

**Arctic and Nordic Seas**

The Arctic improvements are convincing (better Beaufort Gyre, removal of artifacts, improved St. Anna Trough representation). However, the spreading of the Beaufort Gyre into the Canadian Archipelago is "not physical" (p. 10). This should be discussed in more depth — what is the cause (MSS weakness? filtering artifacts?), and how should users treat this feature?

In the Nordic Seas, CLS22 newly resolves the NwAFC along the Mohn Ridge. This is an important advancement and deserves stronger emphasis in the conclusions.

---

## Referee Comment (RC3)

**New Global Mean Dynamic Topography CNES-CLS-22 Combining Drifters, Hydrography Profiles and High Frequency Radar Data**

Solène Jousset1, Sandrine Mulet1, Eric Greiner1, John Wilkin2, Lien Vidar3, Léon Chafik4, Roshin Raj5, Antonio Bonaduce5, Nicolas Picot6, Gérald Dibarboure6

- 5 1CLS, Ramonville Saint Agne, 31250, France

[revised manuscript text omitted]
">https://doi.org/10.24400/527896/A01-2023.003, accessed at <a href="https://https://https://doi.org/10.24400/527896/A01-2023.003">https://https://https://https://https://https://https://https://https://https://https://https://https://https://https://https://https://https://https://https://https://https://https://https://https://https://https://https://https://https://https://https://https://https://https://https://https://https://https://https://https://https://https://https://https://https://https://https://https://https://https://https://https://https://https://https://https://https://https://https://https://https://https://https://https://https://https://https://https://https://https://https://https://https://https://https://https://https://https://https://https://https://https://https://https://https://https://https://https://https://https://https://https://https://https://https://https://https://https://https://https://https://https://https://https://https://https://https://https://https://https://https://https://https://https://https://https://https://https://https://https://https://https://https://https://https://https://https://https://https://https://https://https://https://https://https://https://https://https://https://https://https://https://https://https://https://https://https://https://https://https://https://https://https://https://https://https://https://https://https://https://https://https://https://https://https://https://https://https://https://https://https://https://https://https://https://https://https://https://https://https://https://https://https://https://https://https://https://https://https://https://https://https://https://https://https://https://https://

[revised manuscript text omitted]